# Thermal Stability of Highly Filled Cellulosic Biocomposites Based on Ethylene–Vinyl Acetate Copolymer

**DOI:** 10.3390/polym16152103

**Published:** 2024-07-24

**Authors:** Pavel Gennadievich Shelenkov, Petr Vasilievich Pantyukhov, Svetlana Vladimirovna Aleshinskaya, Alexander Andreevich Maltsev, Zubarzhat Rafisovna Abushakhmanova, Anatoly Anatolievich Popov, Jose Javier Saavedra-Arias, Matheus Poletto

**Affiliations:** 1Department of Biological and Chemical Physics of Polymers, Emanuel Institute of Biochemical Physics, Russian Academy of Sciences, 119334 Moscow, Russia; shell1183@mail.ru (P.G.S.); aam.0205@yandex.ru (A.A.M.); zubarzhat.akh@gmail.com (Z.R.A.); anatoly.popov@mail.ru (A.A.P.); 2Joint Research Center, LLC “Metaclay Research and Development”, 143026 Moscow, Russia; aleshinskaya@gmail.com; 3Higher Engineering School “New Materials and Technologies”, Plekhanov Russian University of Economics, 115054 Moscow, Russia; 4Department of Physics, Universidad Nacional, Heredia 40101, Costa Rica; jose.saavedra.arias@una.ac.cr; 5Postgraduate Program in Engineering of Processes and Technologies, University of Caxias do Sul, Caxias do Sul 95070-560, Brazil; mpolett1@ucs.br

**Keywords:** highly filled biocomposite, ethylene–vinyl acetate copolymer (EVA), wood flour, microcrystalline cellulose, thermal-oxidative stabilization, natural antioxidants

## Abstract

The effect of plant-based fillers on thermal resistance in highly filled biocomposites based on ethylene–vinyl acetate copolymer (EVA) was studied. Wood flour and microcrystalline cellulose were used as fillers. It was shown that the introduction of microcrystalline cellulose into EVA did not affect the thermal stability of the polymer matrix. In contrast, the introduction of wood flour into EVA led to a significant increase in the thermal stability of the entire biocomposite. Oxidation induction time increased from 0 (pure EVA) to 73 min (EVA + wood flour biocomposites). The low-molecular weight phenolic compounds contained in wood flour are likely able to diffuse into the polymer matrix, exerting a stabilizing effect. The discovered stabilizing effect is a positive development for expanding the possibilities of technological processing of biocomposites, including multiple processing.

## 1. Introduction

EVA-based composites, filled with natural dispersed fillers, such as cellulose, starch, and wood flour were characterized in earlier works [1,2,3]. These composites are widely used, including for the creation of biodegradable products. The usage of a natural filler can reduce the cost of a composite based on synthetic plastics [4]. It is known that recycling composites based on synthetic polymers leads to thermal oxidation of the matrix polymer. To prevent thermal-oxidative destruction, antioxidants and heat stabilizers are usually added to the composition. The role of antioxidants (inhibitors) is to break the active chain due to interaction with the peroxide radical [5]. The most widely used synthetic antioxidant is Irganox 1010 (BASF Corporation, Charlotte, NC, USA). However, recent studies showed that synthetic antioxidants were poorly compatible with polymers; they can dissolve in water, diffuse onto the surface of the polymer, and evaporate over an elongated period [6,7]. One of the alternatives is the usage of natural additives as antioxidants and heat stabilizers. Poletto (2020) [8] examined the influence of natural oil additives on the physicochemical parameters and thermal stability of mixtures based on recycled polypropylene with wood flour. It was stated that there is an influence of natural oil additives on the physicochemical parameters and thermal stability of biocomposites based on recycled polypropylene with wood flour. The obtained results confirmed that when mixing wood flour with 2 wt.% of octane oil, the thermal destruction temperature of the polypropylene/wood flour biocomposite increased from 300 °C to 312 °C. At the same time, the strength and flexural modulus of elasticity increased noticeably. That work concluded that natural oils improve interfacial adhesion between wood flour and polypropylene matrix. Another work by Vorobyova and Prykhod (2019) [9] displayed the antioxidant effect of various organic fillers, such as dried and shredded buckwheat husk sowing (*Fagopyrum esculentum*), carposome of crab-of-the-woods (*Laetiporus sulphureus*), carposome of chaga mushroom (*Inonotus obliquus*), and thallus of lichen of oakmoss (*Evernia prunastri*), in composites with low-density polyethylene. The work studied the antioxidant effect of both the fillers themselves and their extracts on the polyethylene matrix. The results showed that the most effective antioxidant was an extract of oakmoss (*E*. *prunastri*), which increased the induction period of polyethylene oxidation by more than 10 times. Cerruti et al. (2009) [10] described the effect of the extract from tomato peel and seeds on the stabilization of polypropylene. The authors argued that lycopene, a carotenoid pigment found in large quantities in tomatoes, is a promising antioxidant for polymers. The effectiveness of other substances contained in plants, including quercetin, α-tocopherol, and cyclodextrin, enhanced polyethylene stabilization; the addition of these substances significantly increased the induction period of the oxidation [11]. The stabilizing effect of flavonoids (chrysin, quercetin, hesperidin, naringin, silibinin) under the influence of UV irradiation and temperature on polypropylene was also studied [12]. In general, a review of the literature data has shown that the use of natural additives as antioxidants in mixtures with synthetic polymers is an effective method of stabilization. However, in a majority of these works, it was not the raw plant particles used, but it was their extracts, where polyphenolic compounds exist in a concentrated form. When a small amount of plant particles is introduced, the effect of thermal stabilization of the polymer matrix is negligible. In highly filled biocomposites (over 50 wt.% of vegetable filler), the effect of thermal stabilization may be more obvious, but this research has not been carried out yet. In previous works devoted to the preparation and investigation of highly filled biocomposites, it was found that the content of VA in the EVA macromolecule had an influence on elongation at the break of the entire biocomposite; the higher the VA content, the higher the elongation [13,14,15]. Also, the effect of thermal stabilization of the polymer matrix due to the introduction of wood flour was accidentally discovered, and our current research investigates this effect more deeply. The use of natural, biodegradable vegetable fillers opens additional advantages, such as thermal stability and waste reduction, for the prospective use of biocomposites. Thus, the main objective of this work was to study the thermal stability of highly filled biocomposites based on wood flour compared to the ones with pure cellulose. It was also important to discover the correlations between the VA content in the EVA macromolecule or molecular mass of the EVA macromolecule and the effect of thermal stabilization by wood flour. For that reason, five different grades of EVA were used.

## 2. Materials and Methods

### 2.1. Materials

Five different grades of EVA produced by LG Chem (Seoul, Republic of Korea), differing in vinyl acetate content and melt flow index, were used as polymer matrices (Table 1).

Wood flour (WF) of deciduous wood provenance, provided by “Novotop” (Moscow, Russia), and microcrystalline cellulose (MCC) grade 101, produced by “Progress” (Kemerovo, Russia), were used as fillers. The chemical composition of the fillers, known from the literature, is presented in Table 2. The fillers were sifted through a sieve with a mesh diameter of 100 microns and dried at 105 °C for 24 h in an oven before mixing with the polymer.

#### Preparation of Biocomposites

Mixing of EVA with natural fillers was performed via heated mixing rolls UBL6175BL (Dongguan, China) with the temperature of the rolls set at 130 °C and 150 °C, and the rotation speed was 8 rpm. As a result, biocomposites based on five grades of EVA with two fillers (MCC and WF) were obtained. The content of the fillers was 50 wt%. The obtained biocomposites were molded by thermohydraulic press GOTECH GT-7014-H30C (Taichung, Taiwan) at 140 °C and 40 kgf/cm^2^ over 1 min. The thickness of the resulting flat sheets varied from 0.4 mm to 0.5 mm.

### 2.2. Methods

(1)Determination of oxidation induction temperature (dynamic OIT)

Tests were carried out following ISO 11357-6:2018 [17]. The sample was heated at a constant rate in an oxygen atmosphere until the oxidation reaction was detected on the thermal curve. The onset of oxidation was indicated by a sharp increase in generated heat, observed via differential scanning calorimeter (DSC). The test was carried out using DSC 214 Polyma NETZSCH (Selb, Germany), at a heating rate of 10 °C/min. The oxygen flow rate was 50 mL/min, standard 40 μL aluminum crucibles without a lid were used, the masses of the samples varied in the range of 10 ± 3 mg, each measurement was carried out at least 2 times, and the average value was used for calculations.

(2)Determination of oxidation induction time (isothermal OIT)

Tests were carried out following the same ISO standard, using the same calorimeter, and crucibles as described above. For isothermal analysis, the sample was heated to 200 °C at 20 °C/min in an inert gas (nitrogen—100 mL/min) until 200 °C, kept for 5 min at this temperature, and then the gas was switched to oxygen (100 mL/min). The test was continued at a constant temperature of 200 °C until the oxidation reaction occurred.

(3)Thermogravimetric analysis (TGA)

The thermal degradation was studied using a thermogravimetric analyzer TGA/DSC3+ Mettler Toledo (Greifensee, Switzerland) following ISO 11358-1:2022 [18]. About 25–30 mg of the crushed sample was placed into a 150 μL crucible made of aluminum oxide. The measurements were carried out in closed crucibles, and the crucibles’ lids had a hole (made by the manufacturer). The measurement was carried out in atmospheric air (100.0 mL/min): 15 min at 30 °C, then heated from 30 °C to 850 °C at a rate of 20.00 K/min.

(4)Fourier-transform infrared spectroscopy (FTIR)

To assess the effect of vinyl acetate content on the interaction of natural filler with EVA, studies were carried out using an FT-803 Simex IR-Fourier spectrometer (Novosibirsk, Russia) with diamond crystal, employing the method of attenuated total reflectance (ATR). The temperature was 23 ± 2 °C, and the wavelength range was 4000 ≤ V ≤ 600 cm^−1^. The change in the peaks for EVA + WF biocomposites was recorded in the range of 1600–1650 cm^−1^ (peaks 1590 cm^−1^ and 1650 cm^−1^). It was seen that 1590 cm^−1^ is sensitive to aromatic compounds and phenols from lignin in wood flour. The value 1650 cm^−1^ is the peak of oxidized phenols (benzophenones) and also is the peak of the double bond in vinyl acetate [19].

## 3. Results and Discussion

The results of TGA indicated that biocomposites with WF were more stable under heating than biocomposites with MCC. The TG curves (solid lines) and their first derivatives (DTG curves, dashed lines) for biocomposites based on EVA 19150 are illustrated in Figure 1. A similar nature of the curves was discovered for biocomposites based on all studied EVA grades (presented in Appendix A). There are two peaks in the DTG curve of pure EVA. The first one (371 °C) corresponds to the degradation of the side vinyl acetate chain and the release of acetic acid. The second one (484 °C) characterizes the decomposition of the polymer backbone. Although biocomposites with WF began to lose weight earlier than the biocomposites with MCC, this decrease was insignificant. However, the peak of maximum weight loss (DTG curve) for the biocomposite with WF relative to the biocomposite with MCC was shifted by 16 degrees (from 358 to 374 °C) to a higher temperature. In addition, the DTG peak intensity of the biocomposite with WF was significantly lower than that of the biocomposite with MCC. Further analysis of the 19150-MCC biocomposite demonstrates that at the first stage of degradation, indicated by a peak at 358 °C, the filler was mainly destroyed (the peak for pure MCC was also detected at 358 °C). The weight loss of the biocomposite at 400 °C was 43% with an MCC content of 50%. Thus, at 400 °C, almost the entire MCC was destroyed. The biocomposite 19150-WF at 400 °C lost 35% of its weight, which indicated that only 15% of the filler remained. Although biocomposites with WF began to lose weight earlier than biocomposites with MCC, they better resisted thermal destruction at higher temperatures.

Table 3 demonstrates the temperatures of thermal destruction of biocomposites on polymer matrices with different vinyl acetate contents. For comparison, the values of pure fillers and pure polymer matrices are also given. Biocomposites with WF showed higher thermal stability than biocomposites with MCC with all polymer matrices. All biocomposites with MCC had a peak in the temperature region where MCC destruction occurs (~358 °C), and the peaks of biocomposites with WF were determined by the destruction of matrices, sometimes even shifting them to the right (toward higher temperatures) along the abscissa axis. Thus, it can be hypothesized that WF has a thermal stabilization effect on polymer matrices.

For the confirmation of the effect of thermal stabilization of the polymer matrix uncovered by TGA, additional investigations of biocomposites on thermal oxidation were carried out. The samples were studied using a differential scanning calorimeter under pure oxygen, so the onset temperatures of the thermodegradation processes were lower than in thermogravimetric analysis (which occurred in an air atmosphere). Figure 2 shows the kinetic dependences of the oxidative induction of EVA-MCC biocomposites depending on temperature: 100–0, 50–50, and 0–100 wt.%, using EVA with different vinyl acetate content.

MCC had a significantly longer period of thermal stability than EVA. The onset of thermal oxidation for EVA and the EVA-MCC biocomposite are very close. This result indicates the absence of chemical interaction between EVA and MCC. Therefore, MCC, which is more stable to oxidation, does not impact EVA in their biocomposites. This initial conclusion was confirmed by the study of the molecular structure of EVA-MCC using the FTIR method. The spectra of EVA-MCC biocomposites (Figure 3) are almost identical to the pure polymers. Therefore, it can be concluded that MCC in the EVA matrix is an inert filler. It should be noted that the onset of EVA oxidation shifts to a higher temperature with increasing concentrations of VA; with a content of 15% VA, the temperature of the onset of oxidation was about 200 °C, and with an increase in VA above 19%, oxidation began above 220 °C. These data look contradictory since it was shown [20] that the introduction of acetate functional groups into the polyolefin chain led to a decrease in thermal stability. However, in that work, thermal stability was assessed by the TGA method in an inert atmosphere based on weight loss, and the dynamic OIT method operates with a change in enthalpy. At the onset of thermal oxidation, the mass of the sample may not decrease; it decreases already with the formation of volatile products, so the results obtained by different methods may not converge.

A different effect of the filler on the thermal stability was found in composites based on EVA with WF. WF begins to oxidize at sufficiently low temperatures, in the region of the thermal stability of EVA. This can be explained by the chemical composition of WF. Unlike MCC, in addition to cellulose, WF contains lignin, hemicelluloses, and extractives. Some of them are more thermostable than cellulose (lignin), and others are less thermostable (hemicelluloses). Therefore, oxidation does not start for all wood components at the same time. TGA curves of pure fillers (Figure 4) indicated an earlier onset and later completion (a wider range) of thermal destruction of WF compared to MCC.

Figure 5 presents the results of oxidation induction temperature determination (dynamic OIT) for EVA-WF biocomposites: 100–0, 50–50, and 0–100 wt.% using EVA with a different content of VA. The EVA-WF biocomposite exhibits a thermal stabilization effect. It begins to oxidize much later at higher temperatures than the individual components of the mixture. In terms of the thermal oxidation onset temperature, it significantly exceeded the performance of both EVA and WF. Low-molecular weight substances can diffuse into the EVA matrix from WF at the processing temperature under shear deformation, and this is the underlying reason why they can increase the stability of EVA against oxidation. Previous work showed the possibility of such a diffusion [21]. The model medium was liquid hexadecane (models the Kuhn segment in polyethylene); at 160 °C, polyphenolic substances (270 nm in the UV spectrum) and chlorophyll (410 nm and 670 nm) diffused into hexadecane from plant-based fillers. In another work, it was shown that the addition of plant components containing polyphenolic substances (birch leaves, mixed herbs hay) into a polyethylene matrix inhibits the oxidation process at a melt temperature of 130 °C [22]. In the paper [23], the effect of thermal stabilization of LDPE with the following agricultural waste as fillers was shown: grape pomace waste, wood shavings, turmeric waste, coffee grounds, and orange peel waste. Adding 4 wt.% of grape pomace waste led to an increase in the thermal oxidation temperature to 60 °C. This corresponds to the addition of 1 wt.% of synthetic thermal stabilizer Irganox 1010. At the same time, reprocessing of the composites led to an increased effect of thermal stabilization due to a more complete diffusion of polyphenols from fillers into the polymer.

Probably, during the process of compounding, phenols (including tannins, dihydroquercetin), which are natural antioxidants, diffuse into the polymer melt from wood flour. Diffusing into the polymer matrix, they stabilize it from oxidative degradation. At the same time, the polymer matrix protects plant particles of fillers from contact with atmospheric oxygen. This synergistic effect causes increased thermal stability of biocomposites with wood flour and other plant fillers containing polyphenolic antioxidants.

It is also possible that physicochemical interactions between polar groups of wood and EVA occurs, probably resulting in hydrogen bonds, which may explain the higher thermal stability of the biocomposite.

The slope of the curves for biocomposites (EVA + WF) is less steep than that for pure WF (Figure 5). The tangent of the thermal oxidation curve’s slope characterizes the oxidation rate of pure WF and biocomposites made of EVA with WF. For biocomposites, the tangent slope of the curves for EVA + WF is three-times less than that for pure WF (Table 4). It can be concluded that the diffusion of antioxidants from WF into the polymer matrix of EVA has a significant effect on the inhibition of thermal-oxidative degradation.

With an increase in the amount of VA in EVA, the oxidation rate decreases. This can be explained by the fact that with increasing VA content, the chemical affinity of EVA to the antioxidants contained in WF increases. This leads to better solubility of the antioxidants in the polymer matrix, and finally, it leads to a slowdown in oxidation processes. The yield of antioxidants was confirmed by studying the structure of biocomposites using FTIR. Figure 6 shows the IR spectra of EVA + WF biocomposites. In contrast to biocomposites with MCC, in the IR spectra of biocomposites with WF, a peak appears in the region of 1600–1650 cm^−1^ and it increases with increasing VA content in EVA. A more detailed examination of the spectra shows that this peak has two maxima, one of which is numbered (1) 1590 cm^−1^, which duplicates the peak of wood flour, and the second (2) 1650 cm^−1^ is a small peak in the EVA spectrum. The peak at 1590 cm^−1^ indicates aromatic compounds such as phenols from wood flour lignin, while the peak at 1650 cm^−1^ indicates oxidized phenols (benzophenones) and at the same time indicates the peak of the double bond remaining in vinyl acetate. With an increase in the content of VA, low-molecular weight substances that diffused from wood flour are better distributed in the volume and surface layers of the biocomposite (increase in the peak of benzene rings by 1590 cm^−1^), where the oxidation reactions of phenols occur (increase in the peak of the carbonyl group of benzophenones at 1650 cm^−1^).

To confirm the discovered patterns, the thermal stability of highly filled biocomposites was assessed by the OIT method in isothermal mode. Pure EVA begins to oxidize earlier than the biocomposite with WF (Figure 7). Moreover, the filler itself (WF) is not subject to oxidation under these experimental conditions. That is, the degradation of WF occurs at a higher temperature as a result of thermal destruction and not due to oxidation. It is noticeable that biocomposites based on EVA with a lower content of VA (15%) have a shorter period of thermal stability (about 60 min). While for EVA biocomposites with 19–28% VA, this parameter increases to 80 min. This increase in the thermal stability of biocomposites can also be associated with better diffusion and solubility of natural antioxidants in EVA with a high VA content. In addition, it is possible that the formation of hydrogen bonds preferentially occurs at a higher content of polar VA groups.

For EVA-MCC biocomposites, no increase in thermal stability was detected. Figure 8 shows the isothermal OIT curves of biocomposites based on EVA 19150 and MCC. The course of the curves based on other EVA trademarks is identical, so they are not shown in the figure. Table 5 contains the oxidation induction time data of all studied biocomposites. It can be seen that biocomposite EVA + MCC starts to oxidize at the same time as pure EVA. This confirms the assumption that there is no intermolecular interaction between MCC and EVA. Consequently, when EVA is filled with MCC, the copolymer is not stabilized, but when it is filled with WF, thermo-oxidative stabilization of EVA occurs.

## 4. Conclusions

This work examined the influence of two natural fillers, WF and MCC, on the thermal stability of highly filled biocomposites based on EVA. These masterbatches were developed for the purpose of being used as an additive to the matrix of synthetic polymers. When the masterbatch additive is introduced to the main polymer, a biodegradable biocomposite is formed, which significantly reduces the decomposition period of synthetic polymers in the natural environment. The results showed that in EVA biocomposites with MCC, the filler does not have a stabilizing effect on the EVA matrix, and the thermal-oxidative degradation of EVA/MCC biocomposites occurs at the same time when the matrix (EVA) starts oxidizing. In biocomposites with WF, a significant effect of thermal-oxidative stabilization was observed, confirmed in both dynamic and isothermal modes. Moreover, it was found that this effect intensifies with an increase in the content of vinyl acetate in EVA. After addition of WF to EVA 28005, the oxidation induction time increased from 0 (for pure EVA) to 73 min (for EVA + WF biocomposite). The discovered effect of thermal stabilization can be explained by the diffusion of phenolic compounds from wood into the polymer matrix during compounding. Phenolic compounds, as natural antioxidants, can protect the polymer from oxidative degradation. Thus, biocomposites filled with WF are more heat resistant than pure matrix polymer and can withstand more cycles of recycling, while being more biodegradable and cheaper. In previous papers, it was shown that extracts from vegetable fillers might be added to the polyolefin polymer matrix and act as efficient antioxidants. In this work, it was shown that the same effect may be obtained in highly filled biocomposites without additional treatments like extraction. The discovered effect of thermal stabilization is most likely characteristic not only of wood flour, but also of other vegetable fillers that are not chemically purified, which makes such biocomposites more promising.

## Figures and Tables

**Figure 1 polymers-16-02103-f001:**
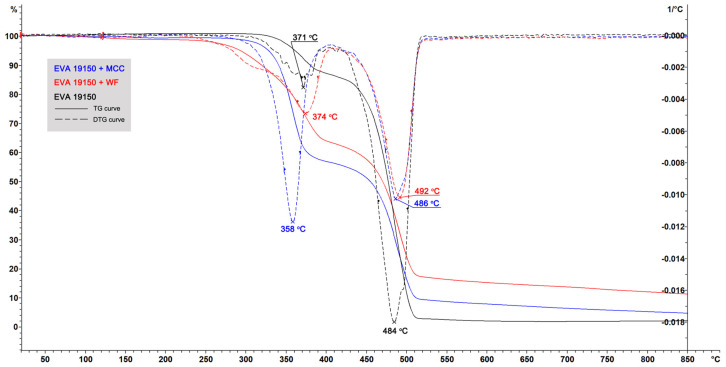
Thermogravimetric analysis (TGA) of biocomposites based on EVA 19150 matrix and pure EVA 19150 matrix.

**Figure 2 polymers-16-02103-f002:**
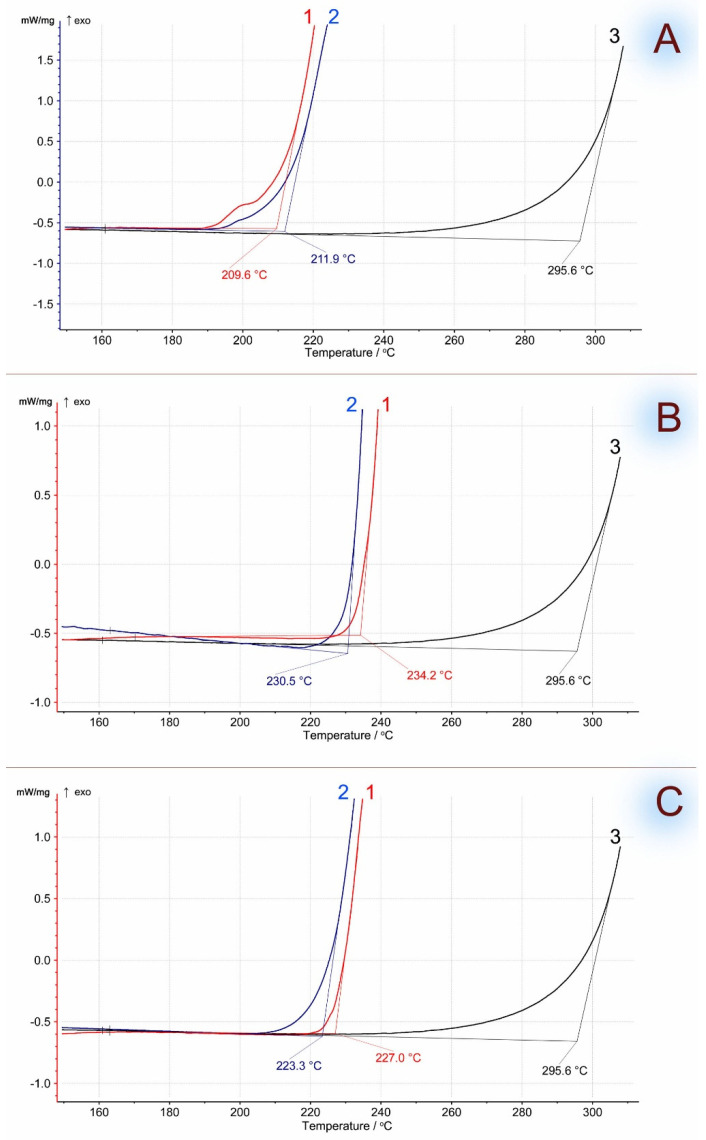
Oxidation induction temperature (dynamic OIT) of MCC-based biocomposites. Curves: 1—EVA, 2—EVA + MCC, 3—MCC. (**A**)—EVA with 15% of VA; (**B**)—EVA with 19% of VA; (**C**)—EVA with 28% of VA.

**Figure 3 polymers-16-02103-f003:**
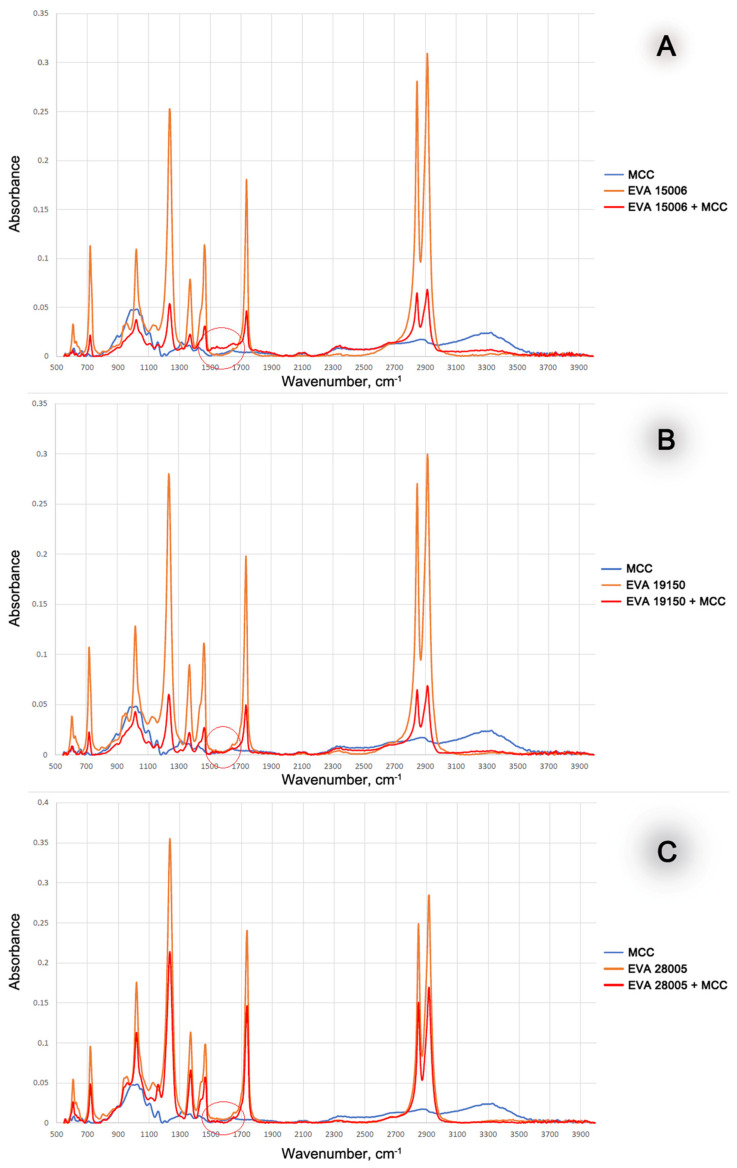
FTIR spectra of MCC-based biocomposites. FTIR spectra of pure EVA and MCC are also presented as a reference. (**A**)—EVA with 15% of VA; (**B**)—EVA with 19% of VA; (**C**)—EVA with 28% of VA.

**Figure 4 polymers-16-02103-f004:**
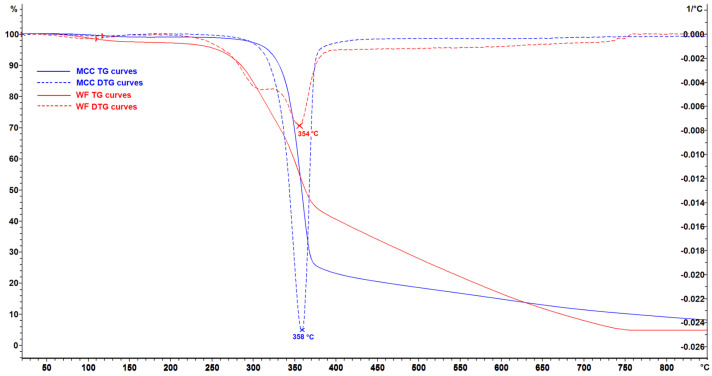
TG (solid) and DTG (dotted) curves of the fillers. Wood flour (WF)—red curves, microcrystalline cellulose (MCC)—blue curves.

**Figure 5 polymers-16-02103-f005:**
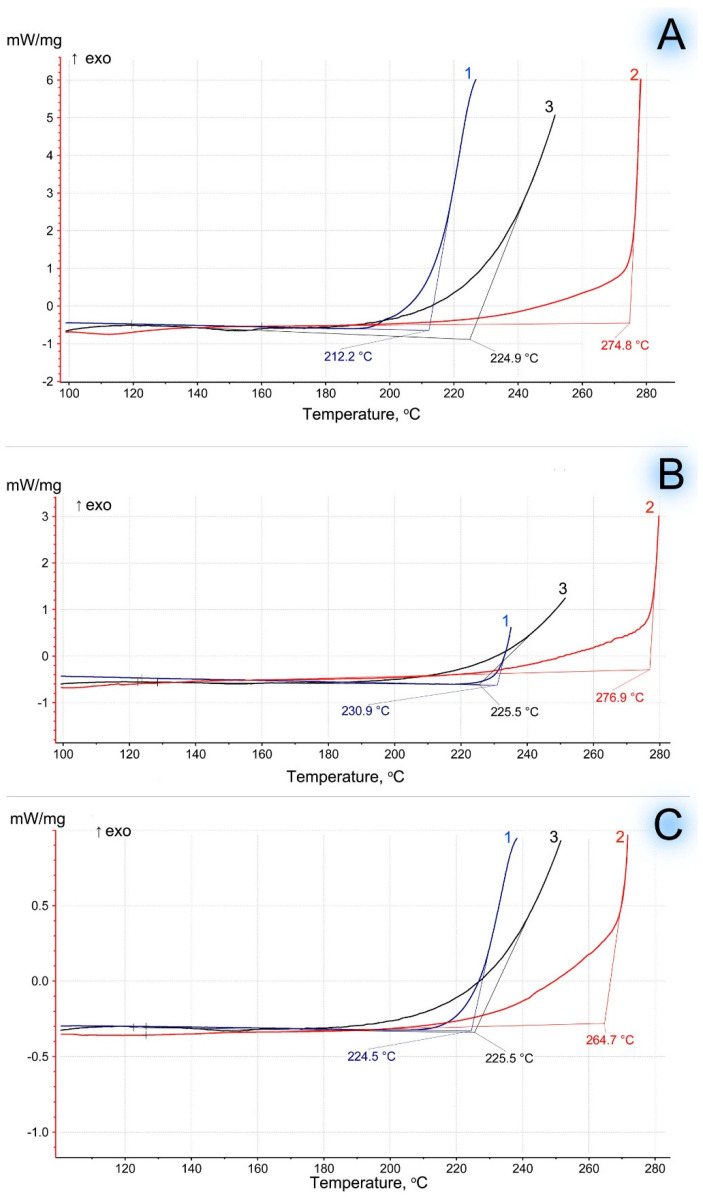
Oxidation induction temperature (dynamic OIT) of WF-based biocomposites. Curves: 1—EVA, 2—EVA + WF, 3—WF. (**A**)—EVA with 15% of VA; (**B**)—EVA with 19% of VA; (**C**)—EVA with 28% of VA.

**Figure 6 polymers-16-02103-f006:**
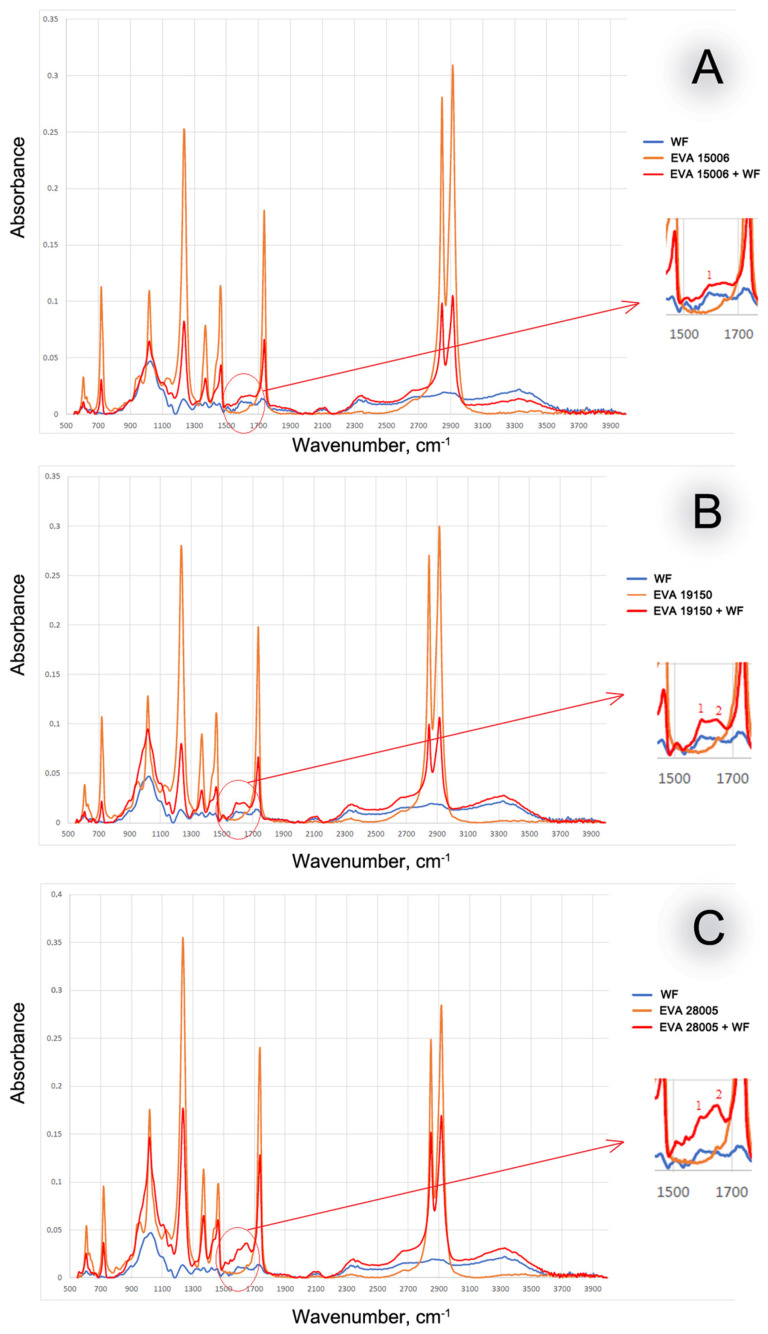
FTIR spectra of WF-based biocomposites. FTIR spectra of pure EVA and WF are also presented as a reference. (**A**)—EVA with 15% of VA; (**B**)—EVA with 19% of VA; (**C**)—EVA with 28% of VA.

**Figure 7 polymers-16-02103-f007:**
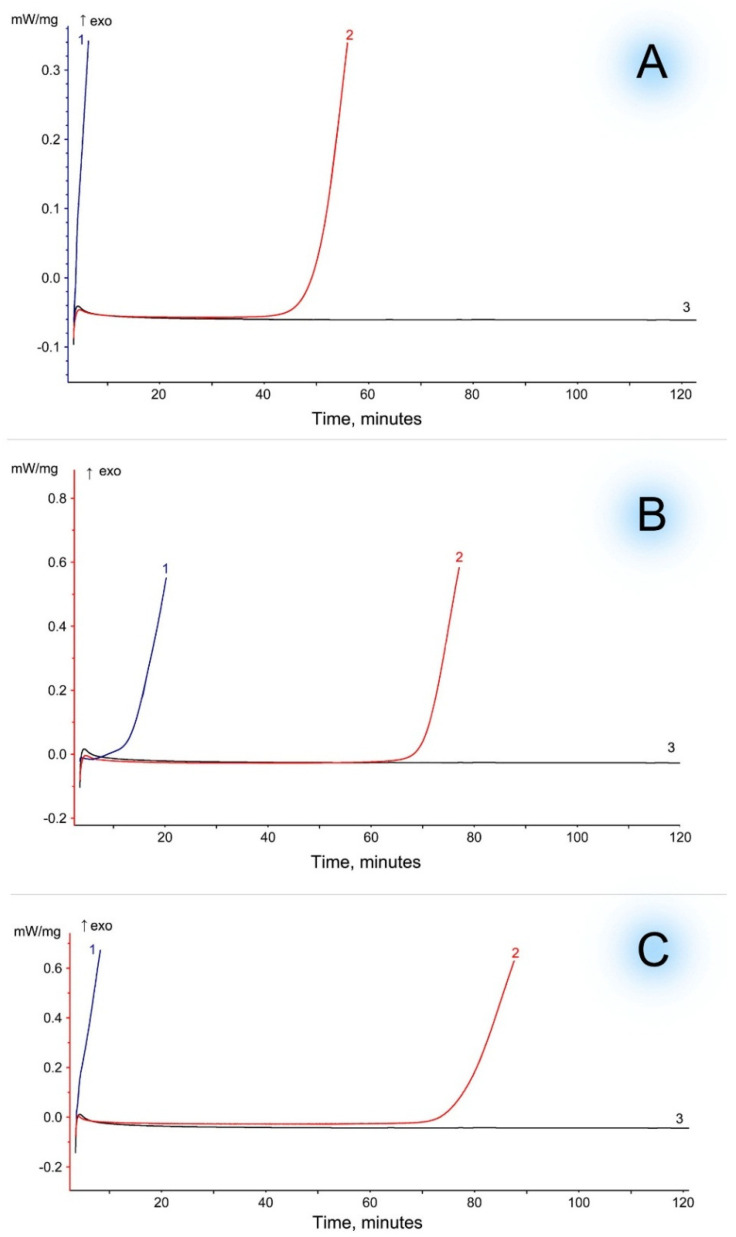
Oxidation induction time (isothermal OIT) of WF-based biocomposites (WF) at 200 °C. Curves: 1—EVA, 2—EVA + WF, 3—WF. (**A**)—EVA with 15% of VA; (**B**)—EVA with 19% of VA; (**C**)—EVA with 28% of VA.

**Figure 8 polymers-16-02103-f008:**
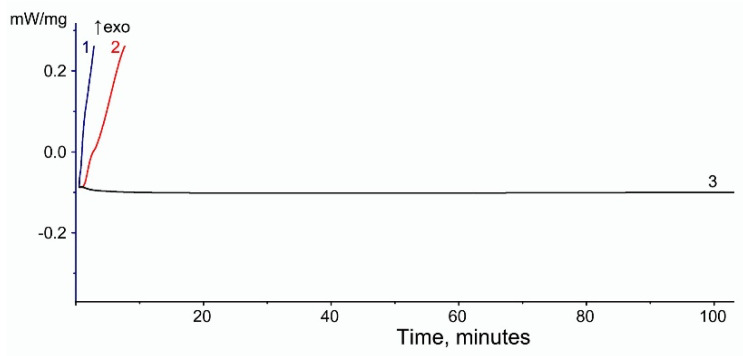
Oxidation induction time (isothermal OIT) of MCC-based biocomposites with EVA 28005 matrix at 200 °C. Curves: (1)—EVA, (2)—EVA + MCC, (3)—MCC.

**Table 1 polymers-16-02103-t001:** Basic characteristics of EVA grades (provided by LG Chem in technical datasheets).

EVA Grade	Content of Vinyl Acetate Groups		Melt Flow Index (MFI) at 190 °C/2.16 kg [g/10 min]
	[wt.%]	[mol.%]	
28005	28	11	5
28025	28	11	25
28150	28	11	150
15006	15	5	6
19150	19	7	150

**Table 2 polymers-16-02103-t002:** Chemical composition of the fillers.

Filler	Cellulose [wt.%]	Lignin [wt.%]	Pentosanes [wt.%]	Polyuronic Acid [wt.%]	Reference
Wood flour (WF)	46	20	29	5	[16]
Microcrystalline cellulose (MCC)	100	-	-	-	-

**Table 3 polymers-16-02103-t003:** Onset, peak, and end temperatures at DTG curves (the first derivatives of TGA).

Composition	T_onset_, °C	T_peak_, °C	T_end_, °C
WF	253	354	756
MCC	311	358	374
EVA 15006	330	374/481	517
EVA 15006 + WF	266	373/492	515
EVA 15006 + MCC	317	359/489	513
EVA 19150	313	372/484	513
EVA 19150 + WF	261	374/492	515
EVA 19150 + MCC	306	358/486	514
EVA 28005	329	367/487	513
EVA 28005 + WF	259	371/488	835
EVA 28005 + MCC	308	356/490	514

**Table 4 polymers-16-02103-t004:** The slope angle of thermal oxidation curves (dynamic OIT).

Composition	Tangent of the Inclination Angle in the Range of 220–240 °C
Wood flour (WF)	0.029 ± 0.001
EVA (15 wt.% of VA) + WF	0.010 ± 0.001
EVA (19 wt.% of VA) + WF	0.009 ± 0.001
EVA (28 wt.% of VA) + WF	0.008 ± 0.001

**Table 5 polymers-16-02103-t005:** Oxidation induction time (isothermal OIT).

Content of VA in EVA, [wt.%]	Filler	Onset Time of Thermal-Oxidative Degradation [min]		
		EVA	Filler	EVA + Filler
15	WF	0.2	>120	46.2
19		9.8		66.8
28		0.2		73.4
15	MCC	0.2	>120	0.2
19		9.8		9.8
28		0.2		0.2

## Data Availability

Data are contained within the article and Appendix A.

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
