# Peer review of "Thermal Stability of Highly Filled Cellulosic Biocomposites Based on Ethylene–Vinyl Acetate Copolymer"

_polymers, 2024, doi:10.3390/polym16152103_

Round 1

Reviewer 1 Report

Comments and Suggestions for Authors

Review Report _ polymers-3025532-peer-review-v1

I have checked the article entitled “Thermal Stability of Highly Filled Cellulosic Biocomposites Based on Ethylene-Vinyl Acetate Copolymer”. The authors need to make the following revision and clarifications before its acceptance. The comments are given below;

1.      The objective of this work has not been mentioned clearly. Please elaborate this point at the end of introduction section.

2.      Why the authors have taken five different grades of EVA for their study? Need to justify this.

3.      Kindly mention the main finding of this work within abstract as well as conclusions section.

4.      Most of the Figures are not clear. Please plot these Figures with good software (like origin).

5.      TGA and DTGA plots have been provided for only EVA 19150 based composites. Please provide the TGA and DTGA plots for other EVA grades-based composites as supplementary figures.

6.      Microstructure of the composites also affect the thermal stability. Authors have not discussed these points.

7.      Which type of application is being expected for these types of bio-composites? Kindly mention this in conclusion section.

8.      EVA shows two stage degradations. 1st stage is due to the degradation of side chain VA and 2nd stage is due to the degradation of main polymer chain. Authors have not discussed this point.

9.      Is the improvement in thermal stability is better compared to earlier study? If so, then please compare it.

Reviewer 2 Report

Comments and Suggestions for Authors

The paper by Pantyukhov and co-authors is devoted to the study of the influence of fillers on the stability of ethylene-vinyl acetate copolymer with different contents of vinyl acetate groups. Using a wide complex of thermoanalytical and spectroscopic equipment, the influence of natural fillers on the thermo-oxidative stability of the obtained biocomposites was determined. The natural filler based on wood flour increased the thermo-oxidative stability several times and showed effective antioxidant properties by interaction with the polymer matrix. The results of this study will optimise the production process and performance properties of biocomposites based on ethylene-vinyl acetate copolymer.

1. It is necessary to specify the material of crucibles used in DSC. In addition, specify the type of crucibles: open or with a pierced lid, also specify the masses of samples measured in TG, DSC, TG/DSC.

2. Table 1. Insert reference of melt flow index values.

3. All TG/DSC curves should be provided in the Supplementary.

4. Heat flow curves for temperature and time should be presented in a more representative form.

5. What is the effect of increasing the content of vinyl acetate groups and fillers on the mechanical properties of biocomposites?

6. The found oxidation onset time for all studied samples should be presented in a table separately.

7. In Figures 7 and 8 it is important to show only the isothermal segment in which the oxidation process is occurring. The beginning of the section should be taken as the beginning of the isothermal segment of the proceeding in the oxygen atmosphere.

We would like to note that a study of the kinetics of the oxidation process under non-isothermal and isothermal conditions could extend the conclusions and draw more attention to this work.

Round 2

Reviewer 1 Report

Comments and Suggestions for Authors

The authors have revised the article as per my given suggestions. Hence, it is recommended to accept this article in its present form.

Reviewer 2 Report

Comments and Suggestions for Authors

The authors of the article have corrected all my comments. I recommend the article for acceptance.